# Group V Chitin Deacetylases Are Responsible for the Structure and Barrier Function of the Gut Peritrophic Matrix in the Chinese Oak Silkworm *Antheraea pernyi*

**DOI:** 10.3390/ijms26010296

**Published:** 2024-12-31

**Authors:** Jing-Wen Tang, Qi Wang, Yun-Min Jiang, Yi-Ren Jiang, Yong Wang, Wei Liu

**Affiliations:** Liaoning Engineering and Technology Research Center for Insect Resources, College of Bioscience and Biotechnology, Shenyang Agricultural University, Shenyang 110866, China; jingwentang307@163.com (J.-W.T.); 2022200057@stu.syau.edu.cn (Q.W.); jiangyunmin544201@163.com (Y.-M.J.); jiangyiren56@syau.edu.cn (Y.-R.J.)

**Keywords:** *Antheraea pernyi*, peritrophic membrane (PM), chitin deacetylase (CDA), microsporidia

## Abstract

Chitin deacetylases (CDAs) are carbohydrate esterases associated with chitin metabolism and the conversion of chitin into chitosan. Studies have demonstrated that chitin deacetylation is essential for chitin organization and compactness and therefore influences the mechanical and permeability properties of chitinous structures, such as the peritrophic membrane (PM) and cuticle. In the present study, two genes (*ApCDA5a* and *ApCDA5b*) encoding CDA protein isoforms were identified and characterized in Chinese oak silkworm (*Antheraea pernyi*) larvae. Although five signature motifs were identified, CDA5 proteins only have the chitin-deacetylated catalytic domain. Spatiotemporal expression pattern analyses revealed that both transcripts presented the highest abundance in the anterior region of the midgut during the feeding period after molting, suggesting their role in chitin turnover and PM assembly. The down-regulation of *ApCDA5a* and *ApCDA5b* via RNA interference (RNAi) was correlated with the breakage of chitin microfibrils in the PM, suggesting that group V CDAs were essential for the growth and assembly of the chitinous layer. Additionally, *ApCDA5a* and *ApCDA5b* may have non-overlapping functions that regulate the morphological characteristics of PM chitin construction in different ways. Larvae injected with double-stranded RNA (dsRNA) against *ApCDA5a* and *ApCDA5b* transcripts were less resistant to infection by *N. pernyi* than those in the control groups. These results revealed that down-regulating *ApCDA5a* and *ApCDA5b* had independent effects on the PM structure and undermined the intactness of the PM, which disrupted the function of the PM against microsporidia infection per os. Our data provide new evidence for differentiating CDA functions among group V CDAs in lepidopteran insects.

## 1. Introduction

The peritrophic membrane (PM) is an acellular structure consisting of chitins, along with structural PM proteins like peritrophins and glycosaminoglycans. This membrane serves as a semipermeable protective barrier that permits the transport of selected materials between the lumen and gut epithelial tissue in arthropods [1]. Most insects exhibit a PM during their developmental stages, which enhances digestive efficiency and decreases pathogen entrance and abrasive fragment injury [2]. The peritrophic matrix acts as a scaffold that is embedded with a variety of digestive enzymes (e.g., amylase, lipases, and serine proteases), and it exhibits strong resistance to endogenous proteases. The composition and organization of the PM may play vital roles in its functions. The main components of the PM are peritrophins, which are also known as invertebrate intestinal mucin, and they cross-link chitin fibrils and maintain the PM structure. They can be divided into several types of chitin-binding proteins that are linked via ionic or covalent interactions and modify the integrity, permeability, and elasticity of the PM [3,4]. *Plasmodium*, *Leishmania* parasites, and viruses have developed strategies to circumvent this barrier, including proteases and chitinases that degrade the PM, thereby facilitating the dispersal of these pathogens to gut epithelial cells [5,6,7,8]. Moreover, chitinolytic enzymes have also been shown to enhance insecticidal activity by disrupting or destroying the PM [9,10]. PM constituents that protect the gut epithelium are potential targets for pathogenic microorganisms because they limit the access to the midgut epithelium.

As the PM serves as a barrier against exogenous pathogens, maintaining its integrity through the regulation of genes encoding components of the PM, such as chitin or peritrophins, is critical. Previous studies on *Spodoptera littoralis* have revealed the importance of peritrophins in the defense against pathogens, including *Bacillus thuringiensis* and nucleopolyhedroviruses [11]. In insects, chitin synthesis, degradation, and modification are regulated to maintain a dynamic balance [12]. Among the enzymes involved in chitin metabolism, chitin deacetylase (CDA), belonging to the carbohydrate esterase family 4 (CE-4), can convert chitin into chitosan by hydrolyzing N-acetylamino linkages, and it is a predominant component of PM [4,13]. Since CDA was purified for the first time from the fungus *Mucor rouxii*, studies have focused on CDAs from various organisms, including plants and insects of various taxa [14]. Although insect CDAs involved in catalytic processes for the deacetylation of chitin substrates have been well characterized, the biological functions of CDAs have only been studied in a few insects. In general, insect CDAs are classified into five orthologous groups according to the presence of conserved domains and sequence similarity [15]. All groups contain polysaccharide deacetylase catalytic domain (CDA). Group I–V CDA-like proteins also have functional motifs of the chitin-binding domain (CBD) and low-density lipoprotein receptor binding domain (LDla). However, members of groups III–IV lack the LDla binding domain and the group V proteins only contain the catalytic CDA domain [16]. Due to the fact that CDAs are distributed widely among the cuticle exoskeleton, PM, and trachea, their synthesis and degradation are associated with growth and development, suggesting that different CDAs serve various functions according to the developmental stages and tissues [17,18].

Most studies have focused on the role of CDAs in cuticular chitin organization, whereas others have investigated gut-specific CDAs of insects. Studies on the CDA gene family of *Tribolium castaneum*, *Locusta migratoria*, and *Leptinotarsa decemlineata* indicate that proper chitin-to-chitosan ratios may be critical for the specific functions of typical chitin-containing tissues, and even for insect survival [19,20,21]. Mun reported that interference with specific CDA isoforms led to the breakage of the chitinous internal tendon cuticle and impaired musculoskeletal connectivity and limb locomotion in *T. castaneum* [22]. Silencing *TcCDA1* or *TcCDA2* also caused considerable molting failure. Moreover, reducing CDAs in the cuticle enhances the sensitivity to pesticides and fungal infections [23], suggesting that the function of these enzymes may be linked to the formation of the cuticle and its barrier function against xenobiotic penetration. Consistent with the spatial expression of CDAs, these esterases have distinct functions in the chitin layers of body parts. Several CDAs in *Bombyx mori*, *Mamestra configurata*, and *Spodoptera exigua* are primarily expressed in the gut and play vital roles in altering the mechanical properties of the PM [4,24,25], indicating that gut-specific CDAs are essential for the growth and structuring of the PM and intestinal wall cell layer. However, the biological properties and specific roles of these CDAs, particularly in the PM barrier function against pathogen penetration, remain elusive. In *Diaphorina citri*, infection with *Escherichia coli* and *Staphylococcus aureus* upregulates the expression levels of *CDA3* in the early and late stages, respectively, suggesting that DcCDA3 might be involved in the immune response of *D. citri* [26]. CDA-like protein is the predominant constituent of the PM in *Ixodes scapularis*. Treatment of ticks with antibodies against IsCDA significantly augmented the pathogen levels of *Borrelia burgdorferi*, which causes Lyme disease [3]. In contrast, the expression of *HaCDA5a* and *5b* was suppressed as early as 12 h and then remained up-regulated after the ingestion of *Bacillus thuringiensis* (*Bt*) toxin for 36 h in the larvae of *Helicoverpa armigera*, indicating that disrupting the link between chitin and the structural proteins of PM leads to the collapse of intestinal defense against pathogens. These findings suggest that gut-specific CDA may be responsible for the homeostasis of PM structure and influence the barrier function that permits the transport of select materials. In this context, gut-specific CDA proteins are thought to play vital yet undefined roles in the intestinal defense system.

The Chinese oak silkworm (*Antheraea pernyi*) is an insect species that is used for silk production and as an edible commodity in China [27]. As a high value-added agricultural product, these moths are widely reared in North China, including Liaoning, Jilin, and Henan Provinces. However, the feeding habits of *A. pernyi* larvae increase their susceptibility to potential pathogens in the wild, resulting in decreased yield and economic losses.

To test this hypothesis, we identified two group V CDAs in *A. pernyi* and examined their transcription patterns in different tissues and larval developmental stages. RNA interference was used to unravel the biological role of gut-specific CDA5a and CDA5b of *A. pernyi*, in forming the PM structure, thereby impeding microsporidium transmission via the alimentary canal. These findings will enrich our understanding of the biological functions of group V CDAs and lay a foundation for further research on the defense systems of physically induced resistance reactions.

## 2. Results

### 2.1. ApCDAs Sequence Analysis

Full-length cDNA sequences of the CDA5a and CDA5b proteins (GenBank accession numbers: OQ446489.1, OQ446490.1) corresponding to the *A. pernyi* CDA gene family, were obtained using primers designed based on a transcriptome database [28]. The predicted properties of the encoded proteins are listed in Table 2. The insect CDAs of group V were identified as the closest relatives through NCBI BLASTp comparisons, which employed the inferred amino acid sequences of each isoform as query sequences. The domain architecture analysis of the two CDA amino acid sequences revealed a polysaccharide deacetylase catalytic domain from family 4 carbohydrate esterases (CE-4) and a signal peptide sequence. Five signature motifs comprising the active sites of the CE-4 family have been described in a previous study [29]. Alignment of multiple CDA sequences from insects and other esterases belonging to CE-4 enzymes, which are known to contain five conserved catalytic motifs, showed that group V CDA lacks some of these residues, with the exception of motifs 1–3, and these substitutions are therefore hypothesized to render the protein enzymatically inactive (Figure 1). Compared with other reported CDA proteins, ApCDA5a and 5b showed higher amino acid sequence similarity (72%) with HaCDA5b and HaCDA5b.

Phylogenetic analysis was performed using orthologs of several insect species based on CDA amino acid residue sequences published in the NCBI GenBank. The insect CDAs were clustered into five groups (I–V) (Figure 2), and ApCDA5a and 5b were closely related to group V CDA sequences from other insects, which clustered distantly from CDA proteins in other groups. It can be inferred that gene differentiation within a group occurs at a later stage and differentiation of genes between groups transpires at an early stage.

### 2.2. Temporal and Spatial Distribution of ApCDA5a and ApCDA5b

The tissue and developmental expression patterns of *ApCDA5a* and *ApCDA5b* were determined using RT-PCR. Fourth instar larvae were selected for tissue distribution analysis because their size is appropriate for the experiments. The results show that *ApCDA5a* and *ApCDA5b* were specifically expressed in the midgut, especially in the anterior region, followed by the middle and posterior midguts (Figure 3A,B). The expression of *ApCDA5a* and *ApCDA5b* was barely detectable in the remaining tissues, except for trace *ApCDA5b* expression in the Malpighian tubules. This is consistent with the findings of abundant expression of *HaCDA5a* in the gut and comparatively low transcription in the Malpighian tubules and fat body [30]. Stage-dependent expression patterns were determined in the midgut of fourth-to-fifth instar larvae, and although the expression was higher during the feeding period, it decreased gradually from L4D7 and partially increased after molting (Figure 4B). The expression levels of *ApCDA5a* and *ApCDA5b* in the larvae were significantly higher than those in the pupae and adults (Figure 4A).

### 2.3. Effect of Nosema Spores on ApCDA Expression

To identify the effects of oral infection with *Nosema pernyi* on the expression of gut-specific *ApCDA5a* and *ApCDA5b*, we examined changes in gene expression in tussah silkworm larvae treated with purified spores of *N. pernyi*. The RT-qPCR results show that the mRNA expression level of *ApCDA5a* was up-regulated as early as 1–9 h in the group fed a diet contaminated with spores and then remained lower. The effect of spore infection on *ApCDA5b* expression was similar to that on *ApCDA5a* expression (Figure 5A,B).

### 2.4. Knockdown of ApCDA5 Interfered with PM Formation

As ApCDA5a and ApCDA5b are expressed mainly in the gut, we sought to determine whether the RNAi-mediated knockdown of ApCDA affects the formation of PM structures and influences the persistence of *N. pernyi* spores within feeding treatments. Thus, double-stranded RNA (dsRNA) for *ApCDA5a* and *ApCDA5b* transcripts (dsApCDA5a/5b) was injected into 1-d-old third instar larvae. The silencing efficiency of dsRNA was verified by profiling the expression after the injection of dsApCDA5a/5b via qPCR. The transcriptional levels of ApCDA5a/5b were substantially lower than those in the control dsRNA targeting the green fluorescent protein (dsGFP) at 24 h after treatment with dsRNA (Figure 6). qPCR analysis results show that the injection of dsRNA can lead to the silencing efficiencies of 65% and 93% in ApCDA5a and ApCDA5b, respectively. However, no observable morphological abnormalities or molting phenotypes were observed after RNAi of gut-specific ApCDA5a/5b in our study. Combining our results with those of related studies, we concluded that group V CDAs did not directly regulate molting in *A. pernyi*.

Next, we assessed whether RNAi-mediated down-regulation of ApCDA5 influences PM formation. Twenty-four hours after the injection of dsApCDA5a/5b, the ultrastructure of the PM of fourth instar larvae was examined using a scanning electron microscope. In the dsGFP-injected larvae, the peritrophic matrix was smooth, tight, and wrinkled and the lamina exhibited a vesicular appearance (Figure 7A,D). This morphology ensures the protective function of PM. However, in dsApCDA5a-treated larvae, the surface of the PM was rough, with irregular pores and large folds (Figure 7B,E). Injection of dsApCDA5b significantly changed the chitin microfibrillar structure of the peritrophic matrix, leading to a loose and disordered reticular conformation of laminar chitin, indicating the breakage of chitin microfibrils in the PM (Figure 7C,F). The exact mechanism responsible for this abnormality is unknown, but observations have indicated that ApCDA5a and ApCDA5b are essential proteins required for chitin matrix organization. The specific functions of ApCDA5a and ApCDA5b may regulate the transcriptional levels of their respective CDA-coding genes. Thus, ApCDA5a and ApCDA5b have non-overlapping functions during chitin matrix construction. RNAi-induced changes in these morphological characteristics may affect PM rigidity and alter the interactions between chitin and its binding proteins.

### 2.5. Functional Analysis of Group V ApCDAs

The resistance of insects to exogenous pathogens was positively related to the structural integrity of the PM. Mature spores of entomopathogenic microsporidia typically penetrated the alimentary tract via the PM. Since ApCDA5 functions in PM construction, we sought to determine whether RNAi-mediated knockdown of ApCDA5a and ApCDA5b influenced the movement of *N. pernyi* spores from the luminal compartment to epithelial cells. The expression of *ribosome*, which is a specific gene of *N. pernyi*, was used to determine the movement of spores outside the lumen. Compared with dsGFP-treated larvae, microsporidium levels exhibited a discernible increase after treatment with dsApCDA5a and dsApCDA5b (Figure 8). Although the groups treated with dsGFP and dsApCDA5a did not significantly differ in their transcripts of *N. pernyi ribosome* 24 h post-infection, we observed significant variations between the groups at 96 h. This difference may account for the different organizing behaviors of CDA5a and CDA5b. This indicates that the weakened ability of dsApCDA5a- and dsApCDA5b-treated larvae to defend against microsporidia infection might lead to the transfer of spores from the intestine to the hemolymph, which is detrimental to the healthy development of tussah silkworms.

## 3. Materials and Methods

### 3.1. Insect Rearing and Infection Experiments

Larvae of *A. pernyi* (strain Xuanda No.1) were obtained from the Research Institute of Tussah, Shenyang Agricultural University (Shenyang, Liaoning Province, China). The rearing temperature was maintained at 25 ± 2 °C with the relative humidity of 60 ± 10 % and a photoperiod of 14 L:10 D [31]. The microsporidium, *Nosema pernyi*, was isolated from infected *A. pernyi* larvae according to procedures as described in a previous study with Percoll (Amersham Pharmacia Biotech, Piscataway, NJ, USA) density gradient centrifugation in the ultracentrifuge (Hitachi, Tokyo, Japan) [32]. The microsporidium suspension of *N. pernyi* was diluted to the desired concentrations with saline and fed to the experimental group of third instar larvae (approximately 1.01 × 10^7^ spores per larva). Each treatment group consists of 30 larvae at least, with 5 replicates at different points in time. Midgut tissue was dissected post-infection within 48 h and stored at −80°.

### 3.2. Identification of CDA5 Genes and Bioinformatic Analysis

The total RNA was extracted using RNAiso Plus kit and quantified by NanoDrop spectrophotometry (Thermo Fisher Scientific, New York, NY, USA) to determine the quality and concentration of the RNA. The first-strand cDNA was synthesized in accordance with the manufacturer’s guidelines in the First-strand cDNA Synthesis Kit (TaKaRa, Dalian, China). The transcriptomic databases of *A. pernyi* [28] were screened for chitin deacetylase genes. RT-PCR was applied to validate the accuracy of the sequences with the primers in Table 1.

Amino acids sequences of *ApCDA5a* and *ApCDA5b* were deduced based on the open reading frame (ORF) sequences, and analyzed on the ExPASy website (https://ca.expasy.org/resources/protparam, accessed on 12 January 2024) [33]. Signal peptides were predicted using the SignalP-5.0 software (https://services.healthtech.dtu.dk/services/SignalP-5.0/, accessed on 12 January 2024). Conserved domains in the CDAs were identified using National Center for Biotechnology Information tools (https://www.ncbi.nlm.nih.gov/cdd, accessed on 18 January 2024). Based on the amino acid sequence of *ApCDA5a* and *ApCDA5b*, a BLAST search was performed on the NCBI website (https://blast.ncbi.nlm.nih.gov/Blast.cgi?PROGRAM=blastp&PAGE_TYPE=BlastSearch&LINK_LOC=blasthome, accessed on 23 January 2024), and the resulting CDA amino acid sequences were downloaded. Clustal X 1.83 was used to perform multi-sequence alignment. MEGA11.0 software was employed for phylogenetic analyses and a phylogenetic tree of insect CDAs was constructed using the neighbor-joining (NJ) method with 1000 repetitions [34]. The tree was displayed with online tools iTOL version 6 [35].

### 3.3. Developmental and Tissue-Specific Expression Profiles of ApCDA5a and ApCDA5b

In order to determine the expression profiles of *ApCDA5a* and *ApCDA5b* during development, a set of gut samples was collected from the fourth and fifth instar larvae for total RNA extraction. Similarly, the tissues including midgut, fat body, epidermis, silk gland, Malpihian tubules, trachea, and genital gland were dissected from the fifth instar day-3 larvae for total RNA extraction. The first-strand cDNA was synthesized as described above and then served as a template for subsequent PCR reactions. The presence and abundance of *ApCDA5a* and *ApCDA5b* during development stages and various larval tissues were examined with semi-quantitative PCR (sqPCR) on S1000 Thermal Cycler (Bio-Rad, Hercules, CA, USA). β-actin was used as the internal reference gene (GenBank accession number: GU176616). Each specimen included five biological replicates. sqPCR reaction was performed under the following conditions: initial denaturation step at 94° for 3 min, followed by 29 cycles of denaturation at 94° for 30 s, annealing at 55° for 30 s, 15 s extension at 72°, final extension at 72° for 5 min. The PCR products were detected with agarose gel electrophoresis and sequenced by Sangon Biotech Co., Ltd. (Shanghai, China).

### 3.4. Quantitative RT-PCR Analysis

Total RNA from the frozen tissues was isolated using the RNAiso Plus kit (Takara, Dalian, China) as described above. The purity and integrity of the RNA samples was examined by spectrophotometer and agarose gel electrophoresis, respectively. First-strand cDNA was generated using Reagent Kit with gDNA Eraser (TaKaRa, Dalian, China). qRT-PCR was performed using the TaKaRa SYBR ExTaq premix reagent in a 20 μL reaction volume on a CFX Connect^™^ Realtime PCR Detection System (Bio-Rad, Hercules, CA, USA), while the qPCR mixture and thermal cycling conditions were set as described in detail in [36]. As the reference gene, the expression of β-actin was used to normalize the target gene. qRT-PCR experiments were performed based on independent RNA sample preparations with three replications and each gene consisted of three technical replicates. The Ct values were determined and used for calculating the relative expression levels of target genes with $2^{-\Delta \Delta Ct}$ method [37]. The specific primers to the examined genes are shown in Table 1. For assessment of *N. pernyi* proliferation levels in response to dsRNA injection, Ribosome transcripts were measured and then normalized to the levels of larvae β-actin transcripts by qPCR.

### 3.5. Functional Analysis of ApCDA5 by RNA Interference (RNAi)

RNA interference was utilized to elucidate the biological functions of gut-specific CDAs in larvae of *A. pernyi*. PCR was performed to generate the cDNA template for the synthesis of double-stranded RNA (dsRNA) for *ApCDA5a* (*ApCDA5a*: 354 bp-767 bp), *ApCDA5b* (*ApCDA5b*: 315 bp-768 bp) and GFP (control). The dsRNA templates were generated using the T7 Quick High Yield RNA synthesis system (New England Biolabs, Ipswich, MA, USA) (Table 2). The synthesized dsRNA sequences were then injected into third instar 1-day larvae (approximately 20 ng per larvae) with a microinjector. The silencing efficiency after injection of dsRNAs for *ApCDA5a* and *ApCDA5b* were quantified by qRT-PCR. The larvae treated with dsRNA injection were maintained separately for phenotypical observation and follow-up study. Each sample contained 5–10 individuals.

For PM compactness tests, the microsporidium suspension of *N. pernyi* was prepared and fed to larvae 24 h post dsRNA injection (approximately 1.01 × 10^7^ spores per larva). RNA samples of the gut were extracted from larvae at various times following feeding, and qPCR analyses were performed to evaluate the *N. pernyi* proliferation levels.

### 3.6. PM Structural Analysis by Scanning Electron Microscopy (SEM)

After 48 h of dsRNA injection, *A. pernyi* larvae were dissected, and PM was rinsed gently with phosphate-buffered saline (PBS) until it became transparent and colorless. Then, the PM was fixed using glutaraldehyde solution at room temperature and stored at 4°. Subsequently, the PM samples were washed three times with PBS (10 min for each time), and then fixed with 1% osmic acid for 2 h at room temperature. After fixation, samples were dehydrated in a series of ascending concentration of alcohol (30, 50, 70, 80, and 90%), with each step lasting 15 min. Then, the samples were dried by CO_2_ critical point and sprayed with platinum film in a vacuum sprayer. Finally, the morphology and structure of the PM were examined and photographed with a scanning electron microscope (Hitachi Science System, Ltd., Tokyo, Japan).

### 3.7. Statistical Analysis

Statistical analyses were conducted using SPSS 21.0 (IBM, Chicago, IL, USA). The data from three biological replicates were expressed as means ± standard deviation. Examining mean disparities across groups using independent samples, Student’s *t*-tests with a *p*-value less than 0.05 were considered significant. The graphs were generated with GraphPad Prism 8 (San Diego, CA, USA).

## 4. Discussion

Several CDAs identified in insect species contain common CDA-like domain sequences, have multiple isoforms, and are essential for chitin organization. CDA-like proteins are clustered into five groups, which diverge from each other not only in the presence of LDLa and ChBD domains but also in their spatiotemporal expression profiles, suggesting different biological functions. Previous studies have demonstrated that CDAs involved in the morphogenesis of cuticle formation affect the organization of cuticular chitin, resulting in the loss of rigidity of the exoskeleton and defects in development [38]. Moreover, the chitin-containing PM plays a critical role in safeguarding insects against pathogens and may physically prevent the passage of pathogens or facilitate protective biochemical interactions. Several group V CDAs in *M. configurata*, *S. exigua*, and *B. mori* are specifically expressed in the gut and associated with the physical characteristics of the PM [13,24,25]. However, details regarding PM formation and whether gut-specific CDAs influence insect biology and PM protective functions in lepidopterans against pathogen infections remain obscure. In this study, we explored the function of group V CDAs associated with the PM in Chinese oak silkworm.

Developmental patterns and tissue-specific expression of different CDA groups in insect species suggest that the chitin deacetylases serve specific purposes in various life stages and tissues. The precise regulation of chitin synthesis, modification, and degradation is vital for insect development. In this paper, two CDA5 genes showed elevated transcription after molting, which resembled the expression profiles of group V CDAs in *B. mori*, *S. exigua*, and *T. castaneum* as the transcripts were expressed highly in the anterior region of the midgut and induced early during the feeding period [1,25,39]. Several integument-specific CDAs have been reported to be induced in the process of metamorphosis and are presumed to be involved in the maintenance of the structural integrity of the cuticular chitin [21,23]. As the PM is turned over during the molting process, we surmise that ApCDA5a and ApCDA5b influence the morphology and physiological properties of the chitin laminae in the PM.

However, why *A. pernyi* has two different CDAs that belong to Group V remains unclear. The alternative splicing of CDA genes was identified in *T. castaneum*, *Apis mellifera*, *L. migratoria*, and *Leptinotarsa decemlineata*; thus, redundancy among CDAs is apparently common among insects [16,20,21,40]. For instance, Luschnig [38] demonstrated that *D. melanogaster* mutant embryos for both *verm* and *serp* encoding similar proteins showed a stronger phenotype than the single mutant embryos, indicating a redundant function of CDA. Silencing either TcCDA2a or TcCDA2b affects cuticle integrity to some content, whereas administration of dsTcCDA2a but not dsTcCDA2b resulted in soft femorotibial joint cuticle and defective locomotion in adults [17,22]. In *L. decemlineata*, knockdown of LdCDA2a and LdCDA2b revealed functional specialization among CDA2 splice variants regarding chitin accumulation [20]. The results indicate that there is no functional compensation for the requirement of either of these proteins by other related CDAs. Here, we show that interfering with chitin deacetylation by down-regulation of specific isoforms belonging to subfamily Group V CDA in *A. pernyi* causes breakage of the chitinous peritrophic matrix. Our observation suggests that these two proteins may have distinct biological roles: ApCDA5b could be required for the laminar compactness of chitin in the PM, whereas ApCDA5a seems to be involved in chitin laminar organization. Notably, the expression levels of CDA5a and CDA5b showed similar increasing trends. The parallel transcriptional profiles of CDA5a and CDA5b in our study resembled those found in previous studies on *H. armigera*. However, the elevated transcription of CDA5 has not been determined previously. Whether the increase occurred as a response to microsporidia penetration into the midgut epithelial cells or as a result of the detection of certain elements that could initiate a response to parasite infection by the host defense system remains obscure. Several studies have revealed the effects of PM-disrupting agents or enzymes in enhancing pathogenic infectivity, including chitinases secreted by baculoviruses, fungi, and bacteria [41,42,43].

Several insect CDA proteins are involved in insect immunity by influencing the mechanical or permeability properties of chitin-containing structures [16]. In *L. migratoria*, the roles of Group I CDAs in cuticle formation have been discussed. LmCDA reduction enhances sensitivity to organophosphorus insecticides, and low LmCDA1 and LmCDA2 expression results in susceptibility to the cuticle barrier function against fungal infection [23]. Increase in CDA expression was observed when fall armyworm and European corn borer larvae were fed transgenic maize producing insect-resistant protease (Mir1-CP), a potent insecticidal protein that attacks the worm PM [44]. Results from such studies indicate that a higher production of PM constituents probably maintains intestine function in a compensatory way. Regulating such enzymes is probably a mechanism used by insects to reduce their susceptibility to ingested toxins. NlCDA3 has been reported to be a gut-specific CDA that is expressed stably at all stages of development, with no observable phenotype or abnormalities in the gut that can be detected following dsRNA injection [45]. In consideration of the absence of PM in *N. lugens*, it is supposed that gut-specific CDAs might contribute to the defense of insects from pathogens by removing the acetyl group of chitin, which may change the properties of the cell walls of fungi.

Microsporidia are obligate intracellular pathogens related to fungi. *Nosema* disease arises from microsporidian parasites, causing devastating economic losses to the silkworm industry, and threatening the conservation and production of *A. pernyi* germplasm [46]. *Nosema* spores infect silkworms through horizontal transmission and primarily damage the intestines, which is the potential cause of poor nutrient absorption in infected silkworms. The PM of *Choristoneura fumiferana* (Lepidoptera: Tortricidae) has been shown to harbor *Nosema fumiferanae* and are confined to the posterior portion of the midgut, which is probably the major site of intrusion [47]. By virtue of the structure and function of the PM in digestion and protection of the overlying epithelium from abrasion and pathogens, PM integrity maintenance through the regulation of genes encoding components of the PM, chitin, and proteins is vital.

Despite evidence that the PM influences disease transmission and impedes the transmission of pathogens in a few arthropods, the physiological functions of the matrix or its role as a barrier in response to invading pathogens are largely uncertain. Zhang [23] and Zhu et al. [48] reported that CDA reduction enhances sensitivity to pesticides and fungal infections. These results are consistent with our current observation that the knockdown of PM-associated CDA affected the PM structure, especially the porosity of the PM, and subsequently affected pathogen persistence. We surmised that RNAi-mediated modifications of the structure, function, or porosity of the PM could facilitate the ability of *N. pernyi* spores to effectively disseminate through the vulnerable gut barrier. Meanwhile, CDA proteins may also interfere with the function of other unidentified proteins crucial for PM formation or function via steric hindrance.

## 5. Conclusions

In summary, the present study provides new insights regarding the functional specialization during chitin matrix construction among gut-specific CDAs in the PM. We observed that CDA reduction enhanced sensitivity to the microsporidia infection. Our results, thus, suggest that the function of group V CDA may be linked to maintaining the structural integrity of the PM that constitutes a physical barrier against microsporidium dissemination and their binding to midgut epithelial cells. However, our data cannot explain precisely how gut-specific CDAs modify the PM to influence defense functions or how the PM limits *N. pernyi* spores within the endoperitrophic lumen. Investigations of the regulatory mechanisms of CDAs and their functions in PM structural formation could help elucidate the biological roles of the associated proteins and enrich our understanding of PM barrier functions against pathogens.

## Figures and Tables

**Figure 1 ijms-26-00296-f001:**
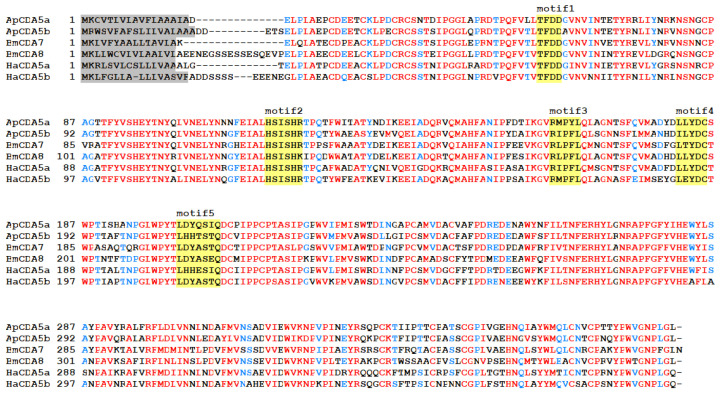
Alignment of inferred amino acid sequences of CDA. The residues that correspond to the consensus residues for the column are highlighted with different colors. The presence of signal peptide residues is indicated by the use of grey highlighting. The five preserved catalytic motifs are highlighted in yellow. The species abbreviations and accession numbers for the sequences included in the alignment are BmCDA7-*Bombyx mori* (XP_004923480.1), BmCDA8-*Bombyx mori* (XP_004923455.1), HaCDA5a- *Helicoverpa armigera* (ADB43612.1), and HaCDA5b- *Helicoverpa armigera* (ADB43611.1).

**Figure 2 ijms-26-00296-f002:**
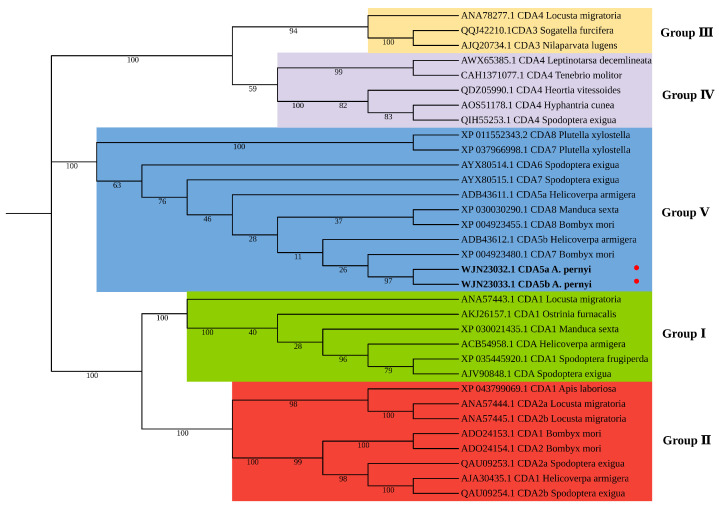
Phylogenetic tree constructed with CDA sequences of *A. pernyi* and other insect species using MEGA 11.0 software with neighbor-joining methods. A bootstrap analysis of 1000 replicates was used. The amino acid residues of CDAs in 18 species were clustered into five major groups.ApCDA5a and ApCDA5b are labeled with red dots.

**Figure 3 ijms-26-00296-f003:**
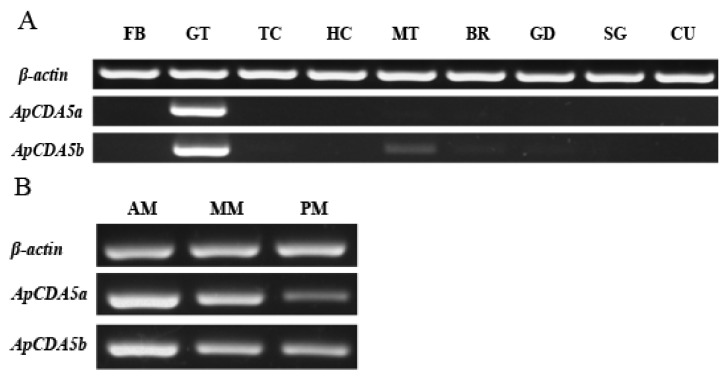
Spatial expression of ApCDA5a/5b in *A. pernyi* larvae. To analyze the expression profiles of *ApCDA5a* and *ApCDA5b*, sqPCR was conducted using total RNA extracted from tissues: (**A**) FB, fat body; MD, midgut; TC, trachea; HM, Hemolymph; MT, Malpighian tubule; BR, brain; GD, genital gland; SG, silk gland; CU, cuticle; (**B**) AM, anterior midgut; MM: middle midgut; PM: posterior midgut. *β-actin* transcript of *A. pernyi* was utilized as an internal reference gene for RT-PCR with the same cDNA template.

**Figure 4 ijms-26-00296-f004:**
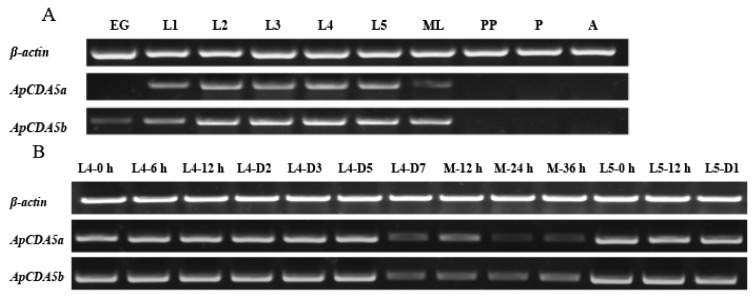
Stage-dependent expression profiles of *ApCDA5a* and *ApCDA5b* during development. (**A**) The temporal expression patterns. EG, embryogenesis; L1-L5, feeding stage larvae of first to fifth instar; ML, mature larva; PP, prepupae; P, pupae; A, adult. (**B**) The expression patterns of ApCDA5a and ApCDA5b in the newly ecdysed larvae between fourth and fifth instar. RT-PCR of *A. pernyi β-actin* transcript with the same cDNA template served as an internal control.

**Figure 5 ijms-26-00296-f005:**
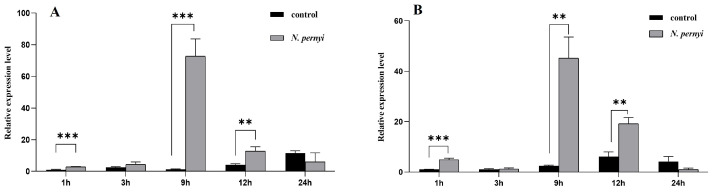
Expression levels of *ApCDA5a* (A) and *ApCDA5b* (B) after *N. pernyi* treatment in *A. pernyi*. Data were standardized using *β-actin* and are provided as the means ± SD of the means from three separate experiments. Statistical analyses were performed using Student’s test. Significant differences are indicated with asterisks. ** *p* < 0.01; *** *p* < 0.001.

**Figure 6 ijms-26-00296-f006:**
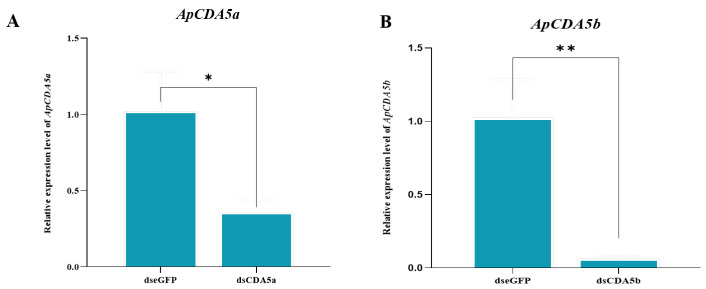
Knockdown of *ApCDA5* transcripts induced by RNA interference. Silencing efficiency of *ApCDA5a* (**A**) and *ApCDA5b* (**B**) after injection of double-stranded RNA targeting the gene of *ApCDA5* (dsApCDA5a/5b) or green fluorescent protein (dsGFP) by qPCR assay. Data are shown in the form of means ± SD from three separate biological replicates. Statistical analyses were conducted with Student’s *t*-test. Asterisks indicate significant differences. ** *p* < 0.01; * *p* < 0.05.

**Figure 7 ijms-26-00296-f007:**
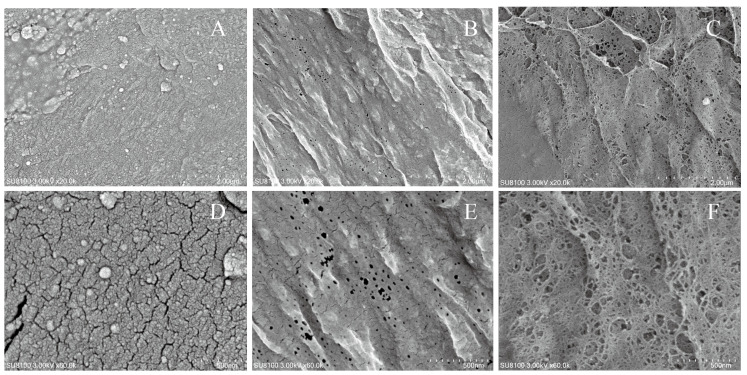
The PM ultrastructure of *A. pernyi* larvae injected with dsRNA. (**A**–**C**) The scanning electron micrographs of the PM at L3D2 after dsGFP (**A**) or dsApCDA5a (**B**) or dsApCDA5b (**C**) injection. (**D**–**F**) is the magnification of (**A**–**C**), respectively. The surface of the PM appears to be normal in larvae of the dsGFP treatment group. Scale bar in (**A**–**C**) is 2 μm, while scale bar in (**D**–**F**) is 500 nm.

**Figure 8 ijms-26-00296-f008:**
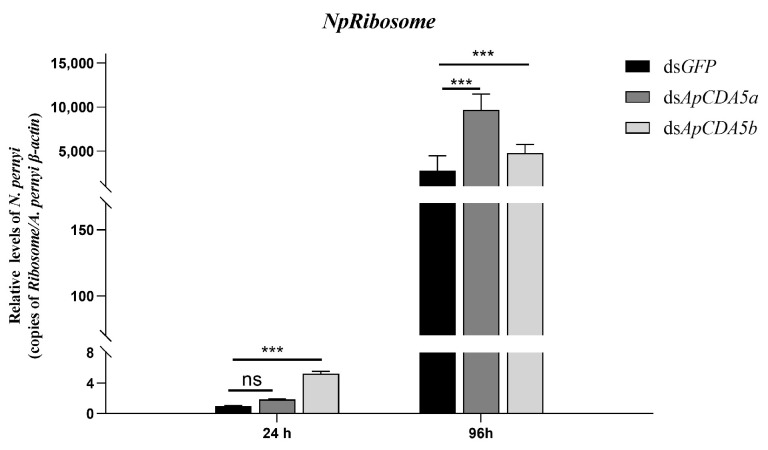
Effects of gut-specific CDAs immunization on *N. pernyi* proliferation and transmission Suspension of *N. pernyi* were fed to larvae injected with ApCDA5a or ApCDA5b dsRNA; then, gut tissue was collected at different points in time. The proliferation of *N. pernyi* was assessed by measuring *Ribosome* transcripts and normalizing to silkworm β-actin using qPCR. Data are represented as the means ± SD from three independent experiments. Statistical analyses were performed using Student’s test. Significant differences are indicated with Asterisks. *** *p* < 0.001; ns: no significant differences.

**Table 1 ijms-26-00296-t001:** Primer sequences used in RT-PCT, qPCR, and dsRNA interference.

Name	Primers
ApCDA5a	Forward: ATTTCCAAAGGATTGCGGTAG
	Reverse: ATAGGACCACAGGAAGTAGCG
ApCDA5b	Forward: TGCCGCTGTAAACGAAATA
	Reverse: ATAATCCAAGAGGGTTTCC
Sq-ApCDA5a	Forward: CTGCCCTACAGCCTCCATTCC
	Reverse: CCACAGGAAGTAGCGGGACAT
Sq-ApCDA5b	Forward: GCTCATTTCGCTAACATTCCC
	Reverse: AACCAGGCATCCTCATCTTCG
Sq-β-actin	Forward: CCAAAGGCCAACAGAGAGAAGA
	Reverse: CAAGAATGAGGGCTGGAAGAGA
qPCR-ApCDA5a	Forward: GCTTACCCAGCCGTTTACAG
	Reverse: CACAGGAAGTAGCGGGACAT
qPCR-ApCDA5b	Forward: GTGCCCAACTGCTTCAATCC
	Reverse: ACCAGGCATCCTCATCTTCG
qPCR-β-actin	Forward: ACCAACTGGGACGACATGGAGAAA
	Reverse: TCTCTCTGTTGGCCTTTGGGTTGA
dsApCDA5a	Forward: TAATACGACTCACTATAGGGCACCACAAACCTTCTGGATAAC
	Reverse: TAATACGACTCACTATAGGGCAGGAATGGAGGCTGTAGGGCA
dsApCDA5b	Forward: TAATACGACTCACTATAGGGACACCGCAGACATACTGGGCT
	Reverse: TAATACGACTCACTATAGGGACCAGGCATCCTCATCTTCGC
dsGFP	Forward: TAATACGACTCACTATAGGGAGATAAACGGCCACAAGTTCAGC
	Reverse: TAATACGACTCACTATAGGGAGAGTGTTCTGCTGGTAGTGGTC

**Table 2 ijms-26-00296-t002:** Sequence characteristics of ApCDA5a and ApCDA5b.

Name	GenBank Accession Number	Protein Length (aa)	Signal Peptide Length (aa)	M. W.(kDa)	pI
ApCDA5a	WJN23032.1	380	17	43.06	4.46
ApCDA5b	WJN23033.1	385	18	43.60	4.44

## Data Availability

Data are contained within the article.

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
