# Peer review of "Group V Chitin Deacetylases Are Responsible for the Structure and Barrier Function of the Gut Peritrophic Matrix in the Chinese Oak Silkworm Antheraea pernyi"

_ijms, 2024, doi:10.3390/ijms26010296_

Round 1

Reviewer 1 Report

Comments and Suggestions for Authors

The manuscript by Tang et al describes two group V chitin deacetylases (CDA5a and 5b), and their potential role maintaining peritrophic membrane (PM) integrity in the gut of the Chinese oak silkworm. Using RNAi-mediated knockdown of each transcript, they noted that reduced CDA transcripts resulted in disrupted PM structure, and that this was correlated with increased susceptibility to the fungal pathogen, Nosema pernyi.

The manuscript was well written, the data were clearly presented, and the conclusions were, in general, justified. There are only a few issues that I would like the authors to address before it is ready for publication:

1. The authors refer to the two proteins, CDA5a and 5b, to be splice variants. If they are splice variants, please show a diagram of the exon/intron arrangement of the two genes. However, given the 70% sequence divergence of the two genes, I doubt they are splice variants, but rather, are protein isoforms encoded by two different genes.

2. Figures 3 and 4 provide a qualitative view of the tissue- and stage-specific expression of the two genes, but a quantitative assessment would have been more informative. Given that the authors have the capability to conducted qRT-PCR, can they explain why they opted for the qualitative approach rather than the quantitative one?

3. Line 338-340.  The conclusion stated in lines 338-340 suggests that there is clear functional difference between the two proteins: “Our observation suggests these two proteins have distinct biological roles: ApCDA5b is required for the laminar compactness of chitin in the PM, whereas ApCDA5a seems to be involved in chitin laminar organization.” This conclusion is based on some visual differences in the panels of Figure 7, but it is unclear how the authors can conclude the precise nature of the disruption to the structural arrangement of chitin macromolecules within the cuticle, based only on these images. The resolution/magnification does not actually allow one to see the organization of the chitin. Ideally, the exact roles of the two CDAs would be resolved by purifying the proteins and examining their impact on chitin using in vitro analyses. I would suggest that the authors add some qualifying words, such as: ApCDA5b “may be” required for the laminar compactness of chitin in the PM, whereas ApCDA5a “could” be involved in chitin laminar organization.

Author Response

Comments 1: The authors refer to the two proteins, CDA5a and 5b, to be splice variants. If they are splice variants, please show a diagram of the exon/intron arrangement of the two genes. However, given the 70% sequence divergence of the two genes, I doubt they are splice variants, but rather, are protein isoforms encoded by two different genes.

Response 1: We sincerely thank the reviewer for careful reading. As suggested by the reviewer, we have corrected the “splice variants” into “two genes encoding CDA protein isoforms” in line 5, and “two transcripts” in line 8.

Comments 2: Figures 3 and 4 provide a qualitative view of the tissue- and stage-specific expression of the two genes, but a quantitative assessment would have been more informative. Given that the authors have the capability to conducted qRT-PCR, can they explain why they opted for the qualitative approach rather than the quantitative one?

Response 2: Thanks for pointing this out. The purpose of Figure 3 and 4 is to illustrate the spatial and temporal specificity of two CDA genes, respectively. And the general expression trends of two genes could be exhibited concisely using the qualitative approach. We agree that quantitative approach would be precisely to detect the difference of gene expression. And qRT-PCR technique will be employed to determine the gene expression levels in follow-up study when necessary.

Comment 3: Line 338-340. The conclusion stated in lines 338-340 suggests that there is clear functional difference between the two proteins: “Our observation suggests these two proteins have distinct biological roles: ApCDA5b is required for the laminar compactness of chitin in the PM, whereas ApCDA5a seems to be involved in chitin laminar organization.” This conclusion is based on some visual differences in the panels of Figure 7, but it is unclear how the authors can conclude the precise nature of the disruption to the structural arrangement of chitin macromolecules within the cuticle, based only on these images. The resolution/magnification does not actually allow one to see the organization of the chitin. Ideally, the exact roles of the two CDAs would be resolved by purifying the proteins and examining their impact on chitin using in vitro analyses. I would suggest that the authors add some qualifying words, such as: ApCDA5b “may be” required for the laminar compactness of chitin in the PM, whereas ApCDA5a “could” be involved in chitin laminar organization.

Response : Thank you for valuable suggestions. We agree with this comment. Therefore, we have made revisions to Conclusion section according to reviewer’s suggestion (line 339-342):

Our observation suggests these two proteins may have distinct biological roles: ApCDA5b could be required for the laminar compactness of chitin in the PM, whereas ApCDA5a seems to be involved in chitin laminar organization.

Reviewer 2 Report

Comments and Suggestions for Authors

This paper looked at two splice variants of the chitin deacetylase gene in the oak silkworm, and used RNA interference to confirm that they are involved in regulating the peritrophic membrane, and that their disruption facilitates infection of the insect with pathogens. Concievably this research could lead to a future development where dsRNA targeting CDAs are added to entomopathogens to make better biopesticides. However, the silkworm in this study is a beneficial, farmed insect, so applications of this work are currently unclear. It's still valid as a basic science paper that helps us understand insect physiology better, given that we are not fully sure how the PM functions in preventing infections, and fits within the scope of this journal.

The introduction is well-arranged and explains the background and importance of studying CDA splice variants, as different CDAs have different functions. As such, it was not known a priori whether the CDAs from this study are in the Pm or, for example, the cuticle, so this research was necessary to confirm hypotheses. I have no changes to request in the introduction.

The methods are thorough, the primer tables is exemplary, and I am satisfied with the breadth of the qPCR expression profiling, RT-PCR, and RNAi. I only have two minor changes, neither scientific:
111 You need to give the full name of the genus Nosema
130-131 Something is off in my pdf and these links extend past the paragraph border, though that's something the layout editor can figure out later.

Results found carbohydrate esterase motifs in the CDA. Is that normal?
The authors confirmed the CDA do not affect molting. the SEMs of the PM are a useful addition.
I am satisfied with the figures.

In the paper, I see references to Group V CDAs and CDA5 genes. I assume CDA5 codes for group V CDAs, but perhaps state the difference or connection between these terms more clearly, or choose between using Group V and 5.

326 delete spaces before commas
328 shouldn't genes like verm and serp be italicized?

Author Response

Comments 1: 111 You need to give the full name of the genus Nosema

Response 1: Thank you for your careful checks. Based on your comments, the full name of genus Nosema has been added at line 111.

Comments 2: 130-131 Something is off in my pdf and these links extend past the paragraph border, though that's something the layout editor can figure out later.

Response 2: We are very sorry for our negligence and the error has been corrected. These links are complete at line 128-130.

Comments 3: Results found carbohydrate esterase motifs in the CDA. Is that normal?

Response 3: CDAs belong to the carbohydrate esterase family 4 and have been well characterized in various species. The five conserved catalytic motifs are usually compared in sequence alignment results.

Comments 4: In the paper, I see references to Group V CDAs and CDA5 genes. I assume CDA5 codes for group V CDAs, but perhaps state the difference or connection between these terms more clearly, or choose between using Group V and 5.

Response 4: To date, the number of CDAs in insects is four to nine, which are divided into 5 groups (Group â… -â…¤) that differ in domain organization, tissue and developmental stage specificity of expression. In this study, ApCDA5a and ApCDA5b only contain a CDA catalytic domain and belong to the group â…¤ CDAs protein. Their name comes from the sequence similarities to Helicoverpa armigera CDA5. However, the number â…¤ represents the classification. Perhaps the CDA5 gene might not be members of group â…¤ protein in other insects.
